

# Detecting nitrogen oxide emissions in Qatar and quantifying emission factors of gas-fired power plants - A four-years study

**Anthony Rey-Pommier[1,2], Frédéric Chevallier[1], Philippe Ciais[1,2], Jonilda Kushta[2], Theodoros Christoudias[2], I. Safak Bayram[3] and Jean Sciare[2].**

[1] Laboratoire des Sciences du Climat et de l'Environnement, LSCE/IPSL, CEA-CNRS-UVSQ, Université Paris-Saclay, 91198 Gif-sur-Yvette, France
[2] The Cyprus Institute, Climate and Atmosphere Research Center, 2121 Nicosia, Cyprus
[3] Department of Electronic and Electrical Engineering, University of Strathclyde, Glasgow G1 1XW, United Kingdom

**Correspondence:** Anthony Rey-Pommier (anthony.rey-pommier@lsce.ipsl.fr)

**Abstract.** Nitrogen oxides ($NO_x$ = NO + $NO_2$), produced in urban areas and industrial facilities (particularly in fossil fuel-fired power plants), are major sources of air pollutants, with implications for human health, leading local and national authorities to estimate their emissions using inventories. In Qatar, these inventories are not systematically updated, while the country is experiencing fast economic growth. Here, we use spaceborne retrievals of nitrogen dioxide ($NO_2$) columns at high spatial resolution from the TROPOMI instrument to estimate $NO_x$ emissions in Qatar from 2019 to 2022 with a flux-divergence scheme, according to which emissions are calculated as the sum of a transport term and a sink term representing the three-body reaction comprising $NO_2$ and hydroxyl radical (OH). Our results highlight emissions from gas power plants in the north-east of the country, and from the urban area of the capital Doha. The emissions from cement plants in the west and different industrial facilities in the south-east are under-estimated, due to frequent low-quality measurements of $NO_2$ columns in these areas. Our top-down model estimates a weekly cycle with lower emissions on Fridays compared to the rest of the week, which is consistent with social norms in the country, and an annual cycle with mean emissions of 9.56 kt per month for the four-year period. These monthly emissions differ from CAMS-GLOB-ANT_v5.3 and EDGARv6.1 global inventories, for which the annual cycle is less marked and the average emissions are respectively 1.44 and 1.68 times higher. Our emission estimates are correlated with local electricity generation, and allow to infer a mean $NO_x$ emission factor of 0.557 $t_{NO_x}.GWh^{-1}$ for the three gas power plants in the Ras Laffan area.

## 1 Introduction

Nitrogen oxides are reactive trace gases that can be converted into other chemical species, including ozone and fine particulate matter. Emissions of $NO_x$ can originate from natural sources (from fires, lightning and soils), but the majority originates from anthropogenic sources, such as vehicle engines and heavy industrial facilities like power plants, steel mills and cement kilns (Vuuren, 2011). High levels of $NO_x$ in the troposphere contribute to the formation of acid rain and smog. They also have a significant effect on human health by causing various respiratory diseases (Bovensmann et al., 1999; Burnett et al., 2004; EPA, 2016; He et al., 2020). To limit those impacts, national and regional governments generally enact a series of air pollution control strategies, which generally take the form of proscriptions on certain polluting technologies, with the aim of reducing the concentration of pollutants at the local level to targets that must be reached within a given timeframe. In the Middle East region, such mitigation strategies are quite recent. The region is thus experiencing increasing levels of $NO_x$ pollution from anthropogenic sources (Osipov et al., 2022), with levels that remain high by international standards (Lelieveld et al., 2015). During the last two decades, Qatar experienced a rapid development based on oil and gas, leading to a degradation of air quality (Mansouri Daneshvar, Hussein Abadi, 2017). However, no official report concerning emissions of pollutants has been publicly available in the last 12 years.

Spectrally resolved satellite measurements of solar backscattered UV-Visible radiation enable the quantification of $NO_2$ in the atmosphere. Such measurements have been providing information on the spatial distribution of tropospheric $NO_2$ for more than 20 years, allowing the identification of many $NO_x$ sources globally (Richter, Burrows, 2002; Celarier et al., 2008). In October 2017, the Sentinel-5 Precursor satellite was launched. Its main instrument is the TROPOspheric Monitoring





Instrument (TROPOMI), which provides daily tropospheric $NO_2$ column densities at high spatial resolution with a large swath width. Columns images can then be used to retrieve $NO_x$ emissions with the use of the continuity equation in steady state. This scheme, known as flux-divergence, requires the use of other physical quantities. In this study, we use this method to quantify the emissions in Qatar based on retrievals from 2019 to 2022 and overcome the absence of recent reporting.

This article is organised as follows: Section 2 provides a description of Qatar and its main emission sources. Section 3 provides a description of the datasets used in this study. Section 4 presents the flux-divergence method to infer $NO_x$ emissions at the scale of the country. Section 5 presents the spatial distribution of $NO_x$ emissions and their seasonal variations. It also presents the limitation of our method to infer emissions for areas above which TROPOMI measurements are often of low quality. Section 6 confronts our estimated emissions to local electricity generation and existing global inventories to provide a sectoral approach of $NO_x$ emissions. Section 7 presents the limits and the uncertainties of the model, and Section 8 our concluding remarks. For the purposes of our study, $NO_x$ emissions are expressed as $NO_2$ throughout this article.

## 2 General features of Qatar and overview of reported $NO_x$ emissions in 2007

Qatar is a country with an area of 11,600 $km^2$, located on the northeast coast of the Arabian Peninsula. The country shares its sole terrestrial border with Saudi Arabia in the south, and has maritime boundaries with Bahrain in the west, Iran in the north and the United Arab Emirates in the east. Qatar has the third largest proven natural gas reserves in the world and non-negligible oil reserves (EIA, 2021). Since 1973, oil and gas revenues increased dramatically, making the country the third largest exporter of natural gas (OPEC, 2020). The resulting economic growth raised Qatar to one of the countries with the highest per capita incomes in the world. Between 2001 and 2019, the average GDP growth rate of the country was 9.1 % (World Bank, 2022), driven by the exploitation of the oil and gas fields, which account for 85 % of its exports and over 60 % of its gross domestic product. Because the incomplete combustion of hydrocarbons produces $NO_x$, the exploitation of such oil and gas resources is a source of air pollution: as a consequence, the transport sector is a source of emissions, as well as the power sector, which is dominated in Qatar by gas power plants. Other sectors, such as cement production, also contribute to the total $NO_x$ budget. Little information is available on the country's emissions and the share of these different sectors. The last official communication dates back to 2011 for emissions in 2007, and uses the IPCC Common Reporting Format sector classification (UNFCCC, 2011). It estimates total emissions of 163 kt (assumed to be expressed as $NO_2$), with the following shares:

- **Power generation**: There are several gas-fired power plants in Qatar which, until the end of 2022, provided all the electricity in the country, with a generation that increased from 4.5 TWh in 1990 to 48 TWh in 2021 (EIA, 2022). According to the report for emissions in 2007, they produced 32.03 kt of $NO_x$, which corresponded to 19.7 % of the country's total emissions. The report indicates an emission factor equal to 0.109 $t_{NO_x}.TJ^{-1}$ or 0.392 $t_{NO_x}.GWh^{-1}$, which is a low but realistic emission factor for an electric mix dominated by gas. More recent data on $NO_x$ emissions from the power sector are not available.

- **Cement production**: The country has three cement production sites. According to the report for emissions in 2007, cement production was responsible for 23.4 % of the country's total $NO_x$ emissions. It should be noted that due to the high temperatures in summer, outdoor activities at the productions sites are reduced, which can result in lower activity between June and September.

- **Road transport**: According to the report for emissions in 2007, the road transport sector accounted for 22.7 % of the country's total $NO_x$ emissions (27.6 % for the total transport sector). Fuel sales do not seem to have any annual seasonality (Al-Attiyah Foundation, 2018): the corresponding $NO_x$ emissions are therefore not variable with seasons. However, they may change gradually with the nature of the vehicle fleet, which includes a growing number of diesel vehicles.

- **Manufacture of Solid Fuels and Other Energy Industries**: This sector includes upstream oil and gas activities and processing operations. Most of historical upstream operations took place in the Dukhan field, located in the west of the country, but the majority of the extraction currently takes place offshore in the North Field, which is one of the largest non-associated gas fields in the world. According to the report for emissions in 2007, these activities accounted for 25.7 % of the country's $NO_x$ emissions.

Figure 1 shows the locations of the industrial facilities in Qatar which are likely to emit significant amounts of $NO_x$. Four regions can be distinguished. The west of the country comprises two large-capacity cement plants (7.0 $Mt.yr^{-1}$ and 5.0 $Mt.yr^{-1}$). The south-east includes a gas power plant, a cement plant and an aluminium smelter, while the north-east comprises the industrial area of Ras Laffan, including three gas power plants. Finally, the urban area of Doha, in the centre-east of the country, concentrates the majority of the population, as well as five gas power plants.



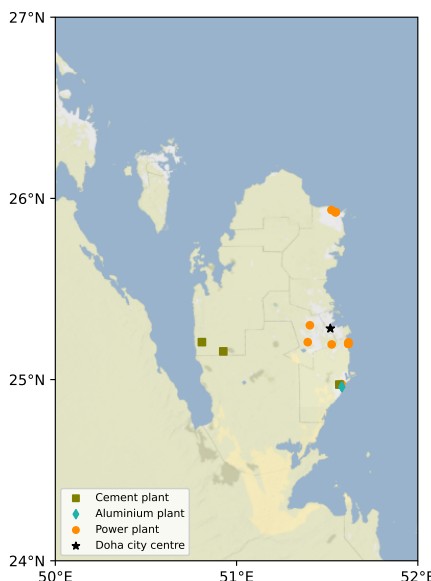

Figure 1: Location of the main industrial facilities (cement kilns, power plants and aluminium smelters) in Qatar. Urban areas are displayed in gray and Doha's city centre is denoted with a star. Industrial facilities and cities in other countries are not displayed. Map tiles by Stamen Design under CC BY 3.0. Data © OpenStreetMap contributors.

## 3   Instruments and data

### 3.1 TROPOMI NO$_2$ retrievals

NO$_2$ can be observed from space with satellite instruments based on its strong absorption features in the 400–465 nm wavelength region (Vandaele et al., 1998). By comparing observed spectra with a reference spectrum, the amount of NO$_2$ in a portion of the atmosphere between the instrument and the surface can be derived. The TROPOspheric Monitoring Instrument (TROPOMI), onboard the European Space Agency's (ESA) Sentinel-5 Precursor (S-5P) satellite, is one of those instruments, providing daily measurements of NO$_2$ around 13:30 local time (LT). Its high spatial resolution (originally 3.5×7 km$^2$ at nadir, improved to 3.5×5.5 km$^2$ since 6 August 2019) allows to observe some of the fine-scale structure of NO$_2$ pollution, such as within cities (Beirle et al., 2019; Demetillo et al., 2020), or near power plants and industrial facilities (Shikwambana et al., 2020; Saw et al., 2021). Tropospheric vertical column densities (VCDs, or simply "columns") are provided by the algorithm after measurement by the instrument, which represents the vertically integrated number of NO$_2$ molecules per surface unit between the surface and the tropopause. An algorithm also provides an air mass factor (AMF), which is used to convert slant column densities into vertical column densities. This factor depends on many parameters, including the albedo of the viewed surface, the vertical distribution of the absorber and the viewing geometry. It is a source of structural uncertainty in NO$_2$ measurements (Boersma et al., 2004; Lorente et al., 2019), which becomes non-negligible in polluted environments. The large swath width of the instrument (∼2600 km) makes it possible to construct NO$_2$ images of VCDs on large spatial scales. We use TROPOMI NO$_2$ retrievals from 2019 to 2022 (S5P-PAL reprocessed data with processor version 2.3.1 from January 2019 to October 2021, OFFL stream with processor version 2.3.1 from November 2021 to October 2022, and OFFL stream with processor version 2.4.0 from November 2022 to December 2022) over Qatar. The arid climate of the country, which offers a large number of clear-sky days throughout the year, enables the calculation of monthly averages based on multiple observations. Its intensive anthropogenic activity, concentrated on a small number of areas, allows to observe high NO$_2$ concentration patterns. Due to its short lifetime (about 1 to 10 hours), background NO$_2$ levels can be orders of magnitude lower than levels near polluted areas. Pollution patterns are therefore characterised by large signal-to-noise ratios above main emitters. Finally, TROPOMI products provide a quality assurance value $q_a$, which ranges from 0 (no data) to 1 (high-quality data). For our analysis of concentrations, we selected NO$_2$ retrievals with $q_a$ values greater than 0.75, which systematically correspond to clear-sky conditions (Eskes et al., 2022), and gridded these retrievals at a spatial resolution of 0.0625°×0.0625°. From August 2019, this resolution is lower than that of the instrument, thus providing a grid for which NO$_2$ VCDs correspond to one or more measurements. The observed plumes remain correctly resolved.



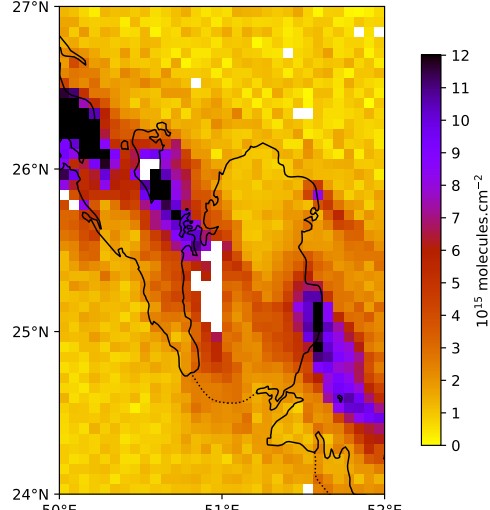

Figure 2: TROPOMI observation of NO$_2$ column densities above Qatar on 12 December 2019. White pixels correspond to areas with low-quality data ($q_a < 0.75$) or no data.

The domain of interest extends between the meridians 50°E and 52°E and between the parallels 24°N and 27°N. TROPOMI tropospheric columns show polluted areas, such as the Ras Laffan area, but also the large urban area of Doha. Observed VCDs are high (5 to $30 \times 10^{15}$ molecules.cm$^{-2}$), which facilitates the observation of NO$_2$ plumes. An example of a TROPOMI image is provided in Figure 2. Within the domain, the meridional component of the wind is generally oriented towards the south. The zonal component varies throughout the year. Strong wind events can cause the pollution from Bahrain to reach the coasts of Qatar. Finally, several areas in the west of the country are not visible by TROPOMI most of the time, because the quality of the corresponding measurements is too low. This phenomenon, which is also present in the east of the country (slightly south of downtown Doha) but with a smaller spatial extent, is studied in details in Section 5.3.

### 3.2 ERA5

The horizontal wind vector field $\mathbf{w} = (u, v)$ is taken from the European Centre for Medium-Range Weather Forecasts (ECMWF) ERA5 data archive (fifth generation of atmospheric reanalyses) at a horizontal resolution of 0.25°×0.25° on 37 pressure levels (Hersbach et al., 2020). The hourly values have been linearly interpolated to the TROPOMI orbit timestamp and re-gridded to a 0.0625°×0.0625° resolution.

### 3.3 CAMS real-time fields

The Copernicus Atmospheric Monitoring Service (CAMS) global near-real-time service provides analyses and forecasts for reactive gases, greenhouse gases and aerosols. Data is gridded on 25 vertical pressure levels with a horizontal resolution of 0.4°×0.4° and a temporal resolution of 3 hours (Huijnen et al., 2019). Here, CAMS concentration fields of hydroxyl radical (OH) are used to calculate NO$_2$ sinks from TROPOMI observations. We also use CAMS temperature field $T$ to account for the the reaction rate for HNO$_3$ production through the calculation of the NO$_2$ lifetime according to Burkholder et al. (2020). The hourly values are also linearly interpolated to the TROPOMI orbit timestamp and re-gridded to a 0.0625°×0.0625° resolution.

### 3.4 Inventories

The Emissions Database for Global Atmospheric Research (EDGARv6.1) and the CAMS global anthropogenic emissions (CAMS-GLOB-ANT_v5.3) are global inventories that provide 0.1°×0.1° gridded emissions for different sectors on a monthly basis. EDGARv6.1 emissions are based on activity data (population, energy production, fossil fuel extraction, industrial processes, agricultural statistics, etc.) derived from the International Energy Agency (IEA) and the Food and Agriculture Organization (FAO), corresponding emission factors, national and regional information on technology mix data and end-of-pipe measurements. The inventory covers the years 1970-2018. CAMS-GLOB-ANT_v5.3 is developed within the framework of the Copernicus Atmospheric Monitoring Service (Granier et al., 2019). For this inventory, NO$_x$ emissions are based on



various sectors in the EDGARv5.0 emissions up to 2015 which are extrapolated to 2021 using sectorial trends from the Community Emissions Data System (CEDS) inventory (Hoesly et al., 2018) up to 2019. From one inventory to another, the names and definitions of the sectors may differ. In EDGARv6.1 and CAMS-GLOB-ANT_v5.3, the emissions for a given country are derived from the type of technologies used, the dependence of emission factors on fuel type, combustion conditions, as well as activity data and low-resolution emission factors (Janssens-Maenhout et al., 2019).

## 3.5 Electricity consumption and production data

As the power sector is one of the main drivers of $NO_x$, we use electricity production and consumption data at several time scales, detailed in Section 5.5. Daily load profiles from February 2016 to January 2017 are taken from Bayram et al. (2018) and used to calculate monthly ratios between the average power demand and the power demand during the overpass of TROPOMI. These ratios are assumed to be valid from 2019 to 2022. We also use monthly electricity generation time series from 2019 to 2022, which are provided by the Qatar Ministry of Development Planning and Statistics (PSA, 2023). From 2019 to 2021, these time series can be completed with monthly reports from Kahramaa, which transmits and distributes the electricity generated by each of the country's power plants with great accuracy (Kahramaa, 2023). The corresponding data for 2022 is not available yet.

## 4   Method

### 4.1 Mask and background removal

In satellite retrievals, the $NO_2$ signal from a sparsely populated area or a small industrial facility may be covered by noise or by the signal generated by natural $NO_x$ emissions. As a consequence, detecting traces of non-natural emissions in TROPOMI $NO_2$ images is not a straightforward process. In the absence of anthropogenic sources, the $NO_2$ columns that are observed constitute a tropospheric background that varies between 0.2 and 1.0 $\times 10^{15}$ molecules.cm$^{-2}$. At the global scale, this background is mostly due to soil emissions in the lower troposphere (Yienger, Levy, 1995; Hoelzemann et al., 2004). In the upper troposphere, $NO_2$ sources include lightning, convective injection and downwelling from the stratosphere (Ehhalt et al., 1992), but the factors controlling the resulting concentrations are poorly understood. According to state-of-art estimates, anthropogenic $NO_x$ accounts for most of the emissions at the global scale, whereas natural emissions from fires, soils and lightning are smaller (Jaeglé et al., 2005; Müller, Stavrakou, 2005; Lin, 2012). In Qatar, the desert climate limits lightning and fire, and the share of the corresponding emissions within the background is thus expected to be low. As a first step, we estimate this background by excluding the part of the domain which is outside the territory of Qatar, as well as other regions with human activity, which include Bahrain and the neighbouring part of Saudi Arabia. The remaining pixels constitute an "external mask" within which we calculate the 5$^{th}$ percentile of the TROPOMI columns. The corresponding value is defined as the tropospheric background. Figure 3 displays the external mask, as well as an "internal mask" that gathers only pixels above Qatar's territory. This external mask is used in Section 5 and Section 6.

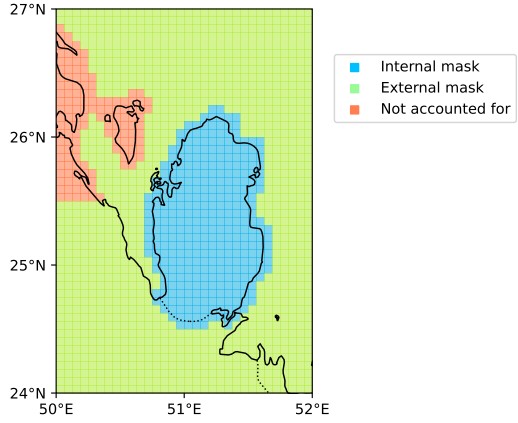

Figure 3: Masks used for $NO_x$ emissions estimates: Green cells are used to estimate the $NO_2$ tropospheric background that is removed from vertical column densities before calculation of emissions (external mask). Blue cells are used to count $NO_x$ emissions attribuable to Qatar (internal mask). Orange cells, which cover urban and industrial areas from other countries, are not considered.





### 4.2 Flux-divergence method

As a second step, we derive top-down $NO_2$ production maps with the flux-divergence method, which consists in applying the continuity equation in steady state as:

$$e_{NO_2} = \mathrm{div}(\Omega_{NO_2}\mathbf{w}) + \Omega_{NO_2}/\tau \qquad (1)$$

The previous equation highlights a transport term $D = \mathrm{div}(\Omega_{NO_2}\mathbf{w})$, obtained by multiplying $NO_2$ VCDs with horizontal wind speeds and the $NO_2$ sinks $S = \Omega_{NO_2}/\tau$. As the local overpass time of TROPOMI is close to 13:30 LT, sinks are dominated by the chemical loss due to the reactions of $NO_2$ with OH (leading to formation of $HNO_3$), which can be described 190 by a first-order time constant $\tau$ as:

$$\tau = \frac{1}{k_{mean}(T, [M]) \cdot [OH]} \qquad (2)$$

Here $k_{mean}$ is the reaction rate that characterises the different reactions between $NO_2$ and OH. Burkholder et al. (2020) provide a general expression of this reaction rate with respect to atmospheric conditions (temperature $T$ and total air concentration $[M]$).

In the atmosphere, OH is a dominant oxidant species. It is mainly produced during daylight hours by interaction between 195 water and atomic oxygen produced by ozone dissociation (Levy, 1971), and its concentration is therefore are strongly correlated with solar ultraviolet radiation (Rohrer, Berresheim, 2006). The direct measurement of OH is possible using spectroscopic methods but the spatial representativeness of the data is limited due to its short lifetime. Most global analyses thus estimate OH budgets from other variable species (Li et al., 2018; Wolfe et al., 2019) with high associated uncertainties (Huijnen et al., 2019). In polluted air, another mechanism for OH production is the reaction between NO and $HO_2$. This reaction, referred 200 to as the $NO_x$ recycling mechanism, illustrates the nonlinear dependence of the OH concentration on $NO_2$ (Valin et al., 2011; Lelieveld et al., 2016). Here, considering OH as the only sink assumes other sinks are negligible. We consider such an approximation as valid (Rey-Pommier et al., 2022). This hypothesis can be validated a posteriori by analysing the calculated $NO_x$ emission maps and verifying the absence of highly negative emissions (which would correspond, all other things being equal, to an underestimation of the sink term) or emissions having the shape of a plume (which would correspond, all other 205 things being equal, to an overestimation of the sink term).

Finally, it should be noted that anthropogenic combustion activities produce mainly NO, which is transformed into $NO_2$ by reaction with ozone $O_3$. $NO_2$ is then photolyzed during the day, reforming NO (Seinfeld, 1989). This photochemical equilibrium between NO and $NO_2$ can be highlighted with the concentration ratio $L = [NO_x]/[NO_2]$. $NO_x$ emissions are therefore obtained by multiplying $NO_2$ production $e_{NO_2}$ by $L$. As diurnal NO concentrations in urban areas are generally 210 above 20 ppb, the characteristic stabilization time of this ratio never exceeds a few minutes (Graedel et al., 1976; Seinfeld and Pandis, 2006). This time being lower than the order of magnitude of the inter-mesh transport time (about 30 min considering the resolution used and the mean wind module in the region), we can reasonably neglect the effect of the stabilization time of the conversion factor on the total composition of the emissions and treat each cell of the grid independently from its direct environment. In Rey-Pommier et al. (2022), this ratio was estimated using the CAMS NO and $NO_2$ concentration fields in 215 Egypt. For Qatar, CAMS data show many outliers for these concentration fields: for some pixels, NO concentrations can be equal to zero. They can be also two times higher than $NO_2$ concentrations in places without any significant feature that would explain it. We therefore do not use these fields and choose a fixed value of 1.32 for this ratio, as used by Beirle et al. (2019). After removal of outliers in the statistics, CAMS values for $L$ suggest that the average value for this ratio ranges between 1.17 and 1.60 with small spatial variations.

## 220 5 Results

### 5.1 Selection of TROPOMI images and averages

We apply Equation 1 to daily TROPOMI images and average the resulting emissions to obtain monthly daytime emissions. This process can be hindered due to the presence of poor quality measurements that prevent the calculation of the divergence in the transport term each day. This happens when a significant part of the country is covered by clouds. Furthermore, the 225 resolution used to grid data ($0.0625°\times0.0625°$, i.e. about $6.3\times6.9$ km$^2$ in this region) is lower than the initial resolution of TROPOMI ($3.5\times7$ km$^2$ until 6 August 2019). For the first eight months of this study, our gridding therefore results in maps having many pixels without data. In the process of estimating mean monthly emissions, we consider these two situations and do not take into account in the averaging the days for which corresponding TROPOMI images have more than 70 % of the territory of Qatar (defined by the internal mask) without data or with poor quality data ($q_a < 0.75$). Furthermore, Bahrain 230 is located 30 km away from Qatar, and the centres of the capitals of the two countries are separated by about 130 km.





Pollution from Bahrain, which is usually transported to Qatar, can reach Doha during strong wind events. In such situations, the errors in ERA5 can alter the estimation of the transport term and thus the $NO_x$ estimates. As a consequence, we also remove days with high wind speeds in the Bahrain/Qatar direction , i.e. days for which the average wind over Bahrain and the marine area between the two countries has a speed higher than 30 km.h$^{-1}$ and an angle between -15° (E$\frac{1}{4}$SE) and -75°

(S$\frac{1}{4}$SE). Figure 4 shows the amount of days considered in the calculation of the average emissions of a given month between 2019 and 2022. It also shows the number of days that have been discarded and the reasons for the corresponding discards.

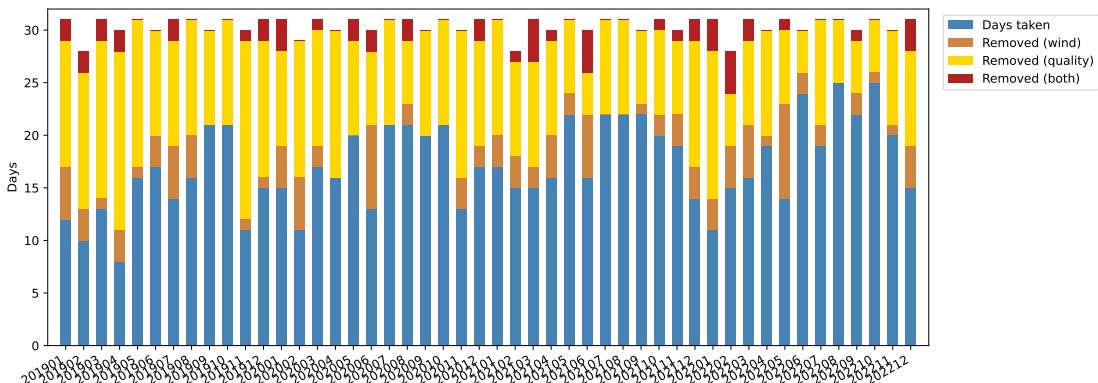

Figure 4: Number of days involved in the mean monthly estimates of $NO_x$ emissions (blue) and days that have been discarded due to strong winds blowing from Bahrain to the south-east (brown), large areas within the internal mask with low-quality data (yellow), or both (red).

Here, monthly average emissions are calculated with between 8 and 25 days of TROPOMI images, with low values for the seven first months of the study due to the lower resolution of TROPOMI during this period. Excluding these months, averages are calculated with 17.9 days. Cases for which too many pixels are not observed by TROPOMI in the internal mask occur

more frequently than cases with strong winds from Bahrain. Overall, we consider most of the inferred monthly averages to be robust. However, at the small scale, there are a small number of pixels with a constant absence of reliable VCD estimations due to low-quality measurements, resulting with pixels having estimations of $NO_x$ with a poor reliability. The treatment of such pixels is precised in Section 5.3.

## 5.2 Spatial distribution of $NO_x$ emissions

We use the flux-divergence model to obtain $NO_x$ emission maps within our domain. For each applicable day (as described in Section 5.1), a background is calculated using the external mask and daily emissions are estimated. These daily emissions are then averaged to obtain a representation of the emissions at 13:30 LT on a monthly scale. For October 2022, a maximum of 25 daily emissions have been average to calculate the corresponding mean monthly emissions map, which is displayed in Figure 5. When the transport term is integrated over large spatial scales, it cancels out due to the mass balance in the

continuity equation between $NO_2$ sources and $NO_2$ sinks. Although the sink term is responsible for most of the total $NO_x$ budget within the domain, the transport term can reach high values at small scale, highlighting hotspots where emissions are concentrated. Such hotspots comprise gas power plants in the north-eastern part of the country and industrial facilities in the south-east. High emissions are also observed in Doha, but in that case, the sink term accounts for most of the $NO_x$ budget due to the spread of sources in the urban area. The west part of the country, where two cement plants are located,

also shows significant emissions. High emissions outside of Qatar are observed in the urban areas of Manama (Bahrain) and Dammam (Saudi Arabia). In the south-eastern corner of the domain, an area of high emissions also seems to stand out from the desert. This area corresponds to a cross-border centre between Saudi Arabia and the United Arab Emirates. It is possible that an important road traffic there is responsible for such emissions, but we have no information to support this hypothesis. Finally, we observe high emissions at sea, up to about 30 km from the east coast of the country. These emissions, of about

$2.0 \times 10^{15}$ molecules.cm$^{-2}$.h$^{-1}$, cannot be solely attributed to shipping activity (Rey-Pommier et al., 2022). Because in this region, the centre of the 40 km × 44 km CAMS pixel is close to the coast, the $NO_2$ lifetime is probably under-estimated. The rest of the domain has relatively low emissions, which is consistent with the absence of major sources of $NO_x$.



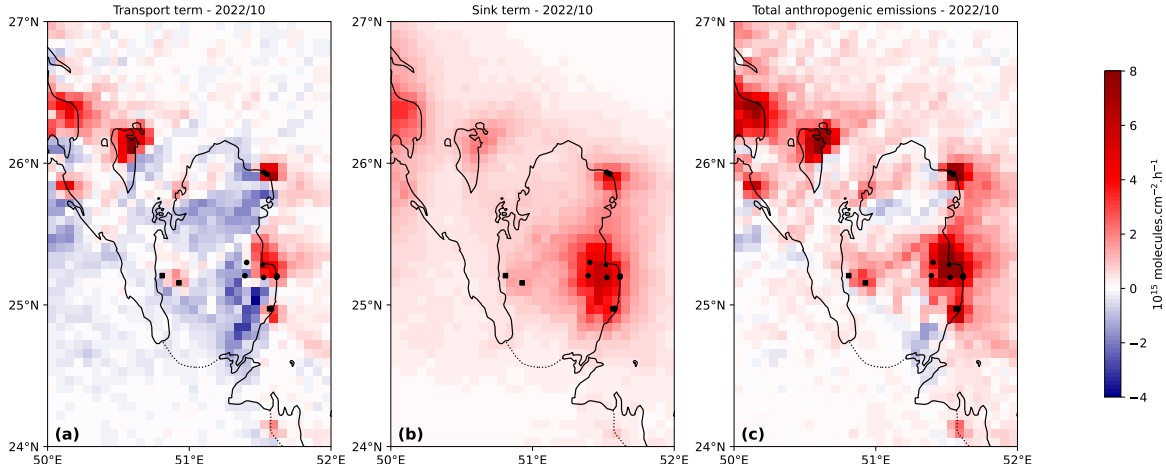

Figure 5: Mean NO$_x$ emissions above Qatar (13:30 LT): transport term (a), sink term (b), and resulting emissions (c) for October 2022. Power plants are denoted with dots, cement plants with squares, and Doha's city centre with a star.

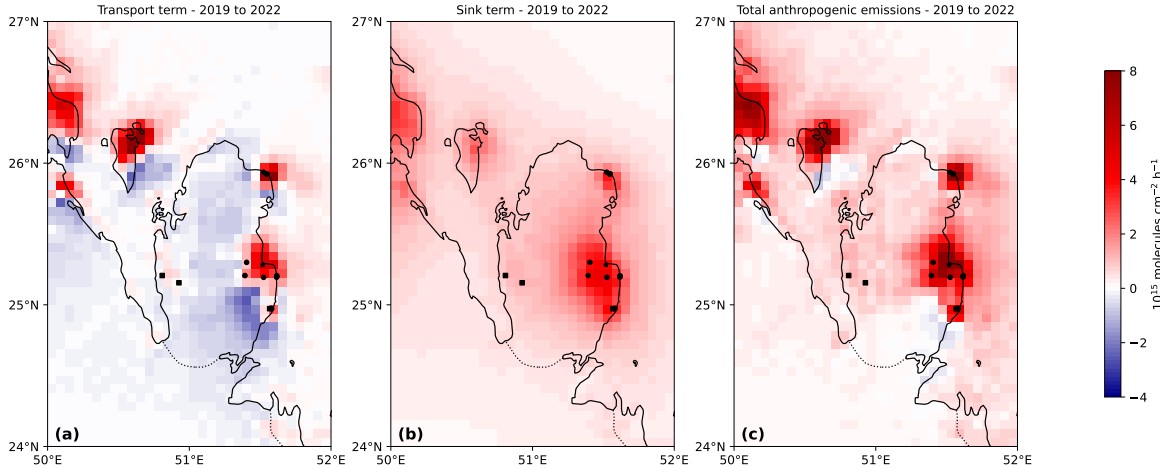

Figure 6: Mean NO$_x$ emissions above Qatar (13:30 LT): transport term (a), sink term (b), and resulting emissions (c) for the 2019-2022 period. Power plants are denoted with dots, cement plants with squares, and Doha's city centre with a star.

Nevertheless, it can be noted that inferred emissions in the domain remain relatively noisy. From monthly maps, we calculate average emissions at 13:30 for the period 2019-2022, which are displayed on Figure 6. The hotspots identified earlier remain visible with a improved signal-to-noise ratio. In particular, emissions in the north-east region are superior to $3.0 \times 10^{15}$ molecules.cm$^{-2}$ for only 7 pixels, which are among the pixels closest to the power plants in the region. In the eastern part of the country, NO$_x$ emissions reflect the main directions of urban expansion in Doha (from the coast towards the west and the southwest), with emissions ranging from $2 \times 10^{15}$ molecules.cm$^{-2}$.h$^{-1}$ to $10 \times 10^{15}$ molecules.cm$^{-2}$.h$^{-1}$. The desert areas display very low emissions (about $0.2 \times 10^{15}$ molecules.cm$^{-2}$.h$^{-1}$), indicating a slight overestimation of the average sink term or an underestimation of the NO$_2$ background. Emissions at sea remain abnormally high on the east coast, but the effect is lower than what is observed on Figure 5. However, unlike for October 2022, for which the emissions from cement plants could be identified, we observe low emissions in the western part of the country.

**5.3 Availability of TROPOMI data above industrial sites**

High-quality measurements by TROPOMI are infrequent above the western and south-eastern part of Qatar, with pixels regularly corresponding to measurements with $q_a < 0.75$. For these pixels, the inferred emissions are only calculated with an



average over a very small number of measurements. In many cases, no measurements are available, and emissions can not be attributed to the pixels. Figure 7 shows the mean TROPOMI coverage over the domain for year 2020 counted as fraction of days with $q_a < 0.75$ within the year. In addition to the west and south-east (corresponding to the southern part of Doha) already identified, the north-east is also affected, but to a lesser extent. This situation is the same for years 2019 and 2021,

and the identified areas have a fraction of observable days lower than 40 %. The situation is different for year 2022, during which the west has a fraction of about 70 % and the south-east 50 % (see Supplementary Material). The effect on total inferred emissions is not negligible, since pixels that are concerned include intensive emitters, such as an industrial area in the south-east and the two large cement plants in the west. For instance, during the 36 months of the period 2019-2021, inferred emissions above these cement plants are lower than $1.5\times10^{15}$ molecules.cm$^{-2}$.h$^{-1}$ for 8 months and greater than this value for

8 months. Emissions could not be inferred for at least one of the plants for the remaining 20 months. Conversely, emissions could be inferred for 10 months out of 12 in 2022, and they were higher than $1.5\times10^{15}$ molecules.cm$^{-2}$.h$^{-1}$ for 6 of those months. In the TROPOMI products, the value of $q_a$ includes automated quality assurance parameters related to different algorithms concerning the presence of clouds, aerosol particles, pressure levels and other physical quantities. For situations with $q_a < 0.75$, retrievals are only sensitive to the NO$_2$ concentrations above the clouds and will depend on model assumptions

for the missing part. They still contain useful information, but this information should be carefully interpreted. Here, all areas that are affected by these low values are located near the coasts. These areas are not characterised by persistent clouds, but they are all located within or close to urban or industrial centres, in which aerosol emissions can be high. However, other regions in Qatar and the neighbouring countries are characterised by high emissions of aerosols without $q_a$ values being particularly high there. For instance, many studies have been providing estimates of NO$_x$ emissions in Riyadh (Beirle et al.,

2019) without indicating anything similar. We observe that these high frequencies of low $q_a$ values are found in several urban and industrial areas on both coasts of the Persian Gulf (especially in Saudi Arabia, United Arab Emirates and Iran). This suggests that the identified persistent low values for $q_a$ come from the calculation of one or several quality assurance parameters. Because the areas displaying low values usually have a rectangular shape, it also suggests a low resolution for these parameters.

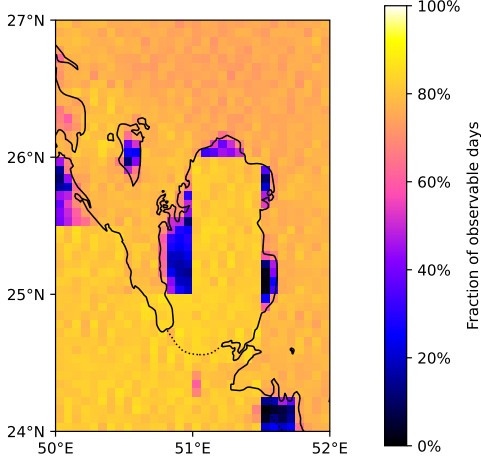

Figure 7: TROPOMI observation density for 2020.

Here, we observe that the use of the OFFL stream (version 2.3.1) from November 2021 to October 2022 generates less pixels for which the value of $q_a$ falls below 0.75 than the use of the S5P-PAL data from January 2019 to October 2021. Furthermore, the last two months of 2022, calculated with the version 2.4.0 of the NO$_2$ product (operational since July 2022), do not show any area with particularly low quality flag values (see Supplementary Material). The latest version of the user manual indicates that in this last version, the surface albedo climatology (SAC) used in the NO$_2$ fitting window and derived from

OMI and GOME-2 was replaced by a SAC derived from TROPOMI observations (Eskes et al., 2022). This new TROPOMI SAC is consistently applied in the cloud fraction, the cloud pressure retrievals, and in the air-mass factor calculation. In the low-$q_a$ areas identified on Figure 7, it is therefore possible that high levels of aerosol-generating industrial activities, in a small number of pixels in the grid, results in a value of $q_a$ below 0.75 in those pixels, but also in all the surrounding pixels due to the low resolution of the parameter responsible for exceeding the threshold, without these pixels being located over particularly

emissive zones. This effect would be absent in the version 2.4.0 of the TROPOMI product. For the first 34 months of our study, the calculated monthly emissions for these areas would be then under-estimaed, because they would correspond to the days with low industrial activity.



To evaluate the impact of this effect, we apply the same model using a threshold value of $q_a$ corresponding to 0.7. This includes many pixels that would be rejected with a threshold of 0.75. As the TROPOMI manual recommends not to use this threshold level, we do not consider these emissions to be representative of human activity in Qatar, and they are not treated as emissions estimates. Nevertheless, it allows us to evaluate the effect of non-detection of cement plants and power plants in the south-east of the domain. The use of a threshold value of $q_a = 0.7$ increases the inferred fluxes by 43.8 % above cement plants in the west, and by 33.4 % above the industrial area in the south-east. Fluxes for the Ras Laffan area in the north and central Doha, which are less affected by the effect described here, are increased by 2.6 % and 3.0 % respectively. It is difficult to determine whether these increases are realistic because they were obtained using columns where $NO_2$ might have been estimated above aerosol and cloud layers. However, they indicate that our model, ran with TROPOMI data before November 2021, leads to a systematic underestimation of $NO_x$ emissions in two of the four most $NO_x$-intensive areas of the country. We note however that on average, these increases are compensated: with a threshold of $q_a = 0.7$, total fluxes within the internal mask are reduced by 2.1 % because fluxes outside the main emissive areas, which comprises most of the mask, are reduced by about 17 %.

### 5.4 Weekly cycle

In Qatar, the official rest day is Friday, and the economic activity of the country is lower during this day than during the other days of the week. We therefore try to characterise this feature, by evaluating the weekly cycle of $NO_x$ emissions. We use the TROPOMI-inferred emissions to obtain averages per day of the week. We use the flux-divergence method to calculate the emissions within the internal mask. We remove the days for which most of the area contains poor quality measurements and those for which the wind from Bahrain blows to the south-east, in the same way as it was conducted in Section 5.1.

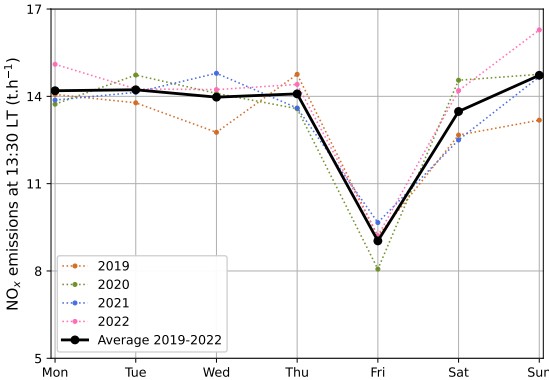

Figure 8: Mean weekly profiles of anthropogenic $NO_x$ emissions at 13:30 LT in Qatar using TROPOMI observations for 2019 to 2022. The average emissions for the four-year period is displayed with the black line.

For a given day, the empty pixels (i.e. those to which it was not possible to assign emissions) are filled with the average of the emissions obtained for the remaining pixels. This choice leads to an underestimation of the emissions when the main emitting areas are covered by clouds or aerosols, and to an overestimation in the opposite case. To limit this effect, we remove from our estimate of annual averages the emissions below the 5[th] percentile and above the 95[th] percentile. Figure 8 shows the resulting daily emissions for the four-year period. Each year, a Friday minimum is observed, defining a weekly cycle. This trend is also observed for mean $NO_2$ column densities. Fridays have average emissions of 9.04 t.h$^{-1}$, which is lower than average emissions for the rest of the week, which reach 14.15 t.h$^{-1}$. In Middle-Eastern countries, this "week-end" effect had already been inferred from satellite measurements by Stavrakou et al. (2020) and Rey-Pommier et al. (2022), and from ground measurements by Butenhoff et al. (2015) on $NO_2$ concentrations, but with a lower difference between Fridays and the other days.

### 5.5 Scaling of emissions and annual cycle

In this study, we estimate the emissions at around 13:30 LT, which corresponds to the moment when TROPOMI overpasses the country. However, anthropogenic activity is not uniform throughout the day, and the emissions inferred from parameters calculated at 13:30 do not correspond to the average emissions of the country. Using the power consumption data from Bayram et al. (2018), Figure 9 shows the mean load curve for the country on April 2016, July 2016, October 2016 and



January 2017. These curves show that the average power injected into the grid is slightly lower than the power injected at 13:30. The ratio between the two powers is minimal in June and reaches 0.911, and maximal in February where it reaches 0.976 (see Supplementary Material). Moreover, according to the traffic congestion index in Doha (Tomtom, 2023), the road traffic around 13:30 shows a congestion peak, slightly lower than the other peaks that occur at the beginning and the end of daytime. This suggests that the average emissions from the road transport sector are also close to those at 13:30, with a similar ratio to that of the power sector.

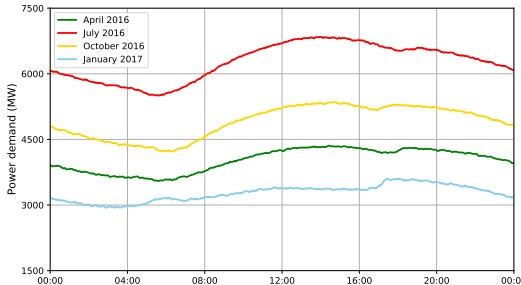

Figure 9: Monthly mean power demand in Qatar for four different months in 2016-2017.

We therefore re-scale the inferred $NO_x$ emissions at the scale of the country by multiplying the monthly estimate for 13:30 with the power ratio to estimate mean monthly emissions. We acknowledge this is a simplification: the power sector has a variable emission factor due to the different technologies in the power plants that are used to respond to the electricity demand over the year, and the road congestion level does not necessarily reflect traffic emissions. Moreover, other sectors, such as cement production, whose seasonality is unknown to us, might behave differently.

Within the internal mask, Figure 10 shows total $NO_x$ emissions in Qatar using our method. Since the $NO_2$ background has been removed from VCD calculations, the total $NO_x$ budget calculated for each month corresponds to the $NO_x$ production by human activities. The annual variability is marked, with higher emissions in summer and lower emissions in winter. On average, $NO_x$ emissions reach 9.56 kt per month. With mean emissions of 9.29 kt and 10.08 kt, 2019 and 2022 appear to be the years with lowest and highest mean emissions respectively. No seasonal cycle seems to appear for VCDs, which suggests more intense oxidation in summer, which is captured by higher OH values for summer months in CAMS data. The uncertainties, detailed in Section 7, are higher for the months for which emissions could not be attributed for the western and south-eastern parts of the territory. This hinders year-on-year comparisons. Consequently, although emissions in the March/April/May (MAM) 2020 period are 96 % and 87 % of the level of emissions in MAM 2019 and MAM 2021 respectively, we can not assert whether this effect can be attributed to the restrictions put in place from March to May 2020 to tackle the Covid-19 pandemic. This caution is all the more necessary as these restrictions have been less stringent than in other countries (Hale et al., 2021). Similarly, higher emissions during 2022 could be due to the different values reached by $q_a$ in the 2.4.0 version of the TROPOMI product with respect to the 2.3.1 version. However, it can not be excluded that these higher emissions were due to a more intense activity in 2022 in the context of organising the FIFA World Cup.

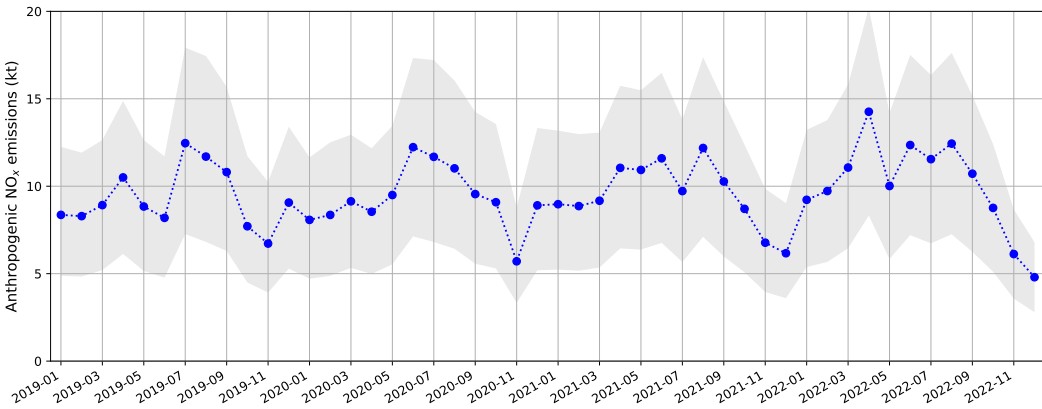

Figure 10: Total TROPOMI-derived anthropogenic $NO_x$ emissions in Qatar from 2019 to 2022.



## 6     Comparison with air pollution inventories and electricity generation data

As Qatar is a small country with an undiversified economy, the annual cycle of the TROPOMI-inferred emissions can be interpreted. Reported emissions reached 163 kt in 2007, but a significant part did not take place within the terrestrial
borders of the country. To compare such reported emissions with those calculated within the internal mask, emissions from upstream oil and gas operations (41.86 kt), which mostly take place offshore, must be removed from the previous budget. We also remove emissions from navigation (4.35 kt), civil aviation (3 kt) and fugitive emissions (4.28 kt). Reported terrestrial emissions therefore reach the approximate value of 109.51 kt for 2007, i.e. an average of 9.13 kt per month, with the main $NO_x$ emitting sectors being power generation, transport and cement. Total emissions, as well as and their share in the total
$NO_x$ budget, may have varied since then. This section confronts our estimated emissions with local electricity generation and existing global inventories in order to provide a sectoral approach of $NO_x$ emissions and their annual variability.

### 6.1 Comparison with electricity generation data

In Qatar, all electricity production comes from gas-fired power plants, with three plants located in the north-east (Ras Laffan area). Other gas power plants are located in the urban area of Doha and in the south-east of the country. Figure 11 shows the
total electricity production and the share of the power plants in Ras Laffan between 2019 and 2021 as provided by Kahramaa. Data regarding the electricity generation by power plant in 2022 is not yet available. The electricity generation is 2 to 2.5 times higher during summer months than during winter months. This increased generation is mostly due to the use of air conditioning (Gastli et al., 2013), which is also captured on Figure 9.

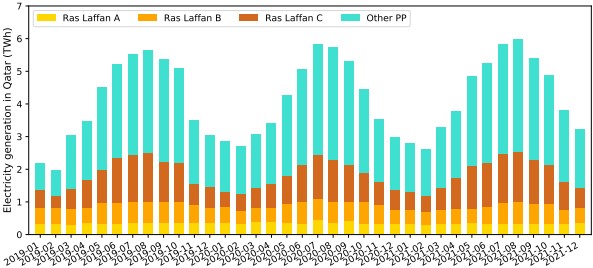

Figure 11: Electricity generation in Qatar from 2019 to 2021 according to the Planning and Statistics Authority reports. Generation from to the three Ras Laffan power plants, in the north-eastern part of the country, are displayed in yellow, orange and brown, while generation from other power plants are displayed in light blue.

The inferred emissions have a behaviour that is similar to electricity generation, with higher emissions in the warmer
390   months. The comparison of the time series of $NO_x$ emissions and the electricity generation data for the four-years period provides a correlation coefficient of $R^2 = 0.400$. This value is relatively low, but it should be noted that monthly emission maps can be very noisy due to averaging over too few days. For instance, on April 2019, the spatial distribution of emissions for this month shows non-negligible emissions in the central part of the country where no significant emitter is located. We can therefore consider that the average is not calculated with enough data to limit noise effects. When we only keep in the analysis
the mean monthly emissions calculated on the basis of more than 18 days (the average over the 4-year period being 17.1), 19 points out of 48 are retained and the correlation is improved, with a coefficient which then reaches $R^2 = 0.657$, with a slope of 1.773 $t_{NO_x}.GWh^{-1}$. This value would be equal to the emission factor of the power sector, provided that it was the only source of variable $NO_x$ emissions during the months that are considered. This value must therefore be considered as an upper limit estimate. Figure 12 shows the comparison between electricity generation and emissions in the two cases. The points
that are retained in the second case mostly correspond to months with high emissions, notably in autumn. Among these 19 points, only 2 correspond to winter or spring months (December to May) whereas 7 correspond to summer months (June to August) and 10 to autumn months (September to November). The improved correlation must be interpreted with caution, but it might be possible that other $NO_x$-producing sectors have a seasonality which is the opposite to that of the power sector. If the transport sector has no particular seasonal cycle, it is possible that industry has a higher production during winter and
spring months, leading to higher industrial $NO_x$ emissions during this period. As mentioned before, mid-day temperatures in summer are high in Qatar, preventing labor in outdoor activities, and this interpretation could be investigated. With the exception of the two cement plants in the west of the country, most of the industries are located in the urban area of Doha or Ras Laffan, where emissions from several sectors overlap over distances smaller than the resolution of TROPOMI.





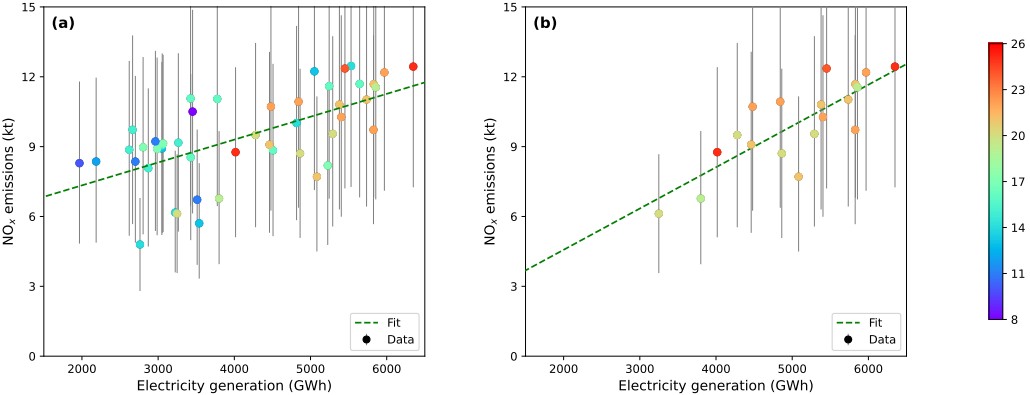

Figure 12: Comparison between monthly TROPOMI-derived NO$_x$ emissions above Qatar and electricity generation according to Planning and Statistics Authority reports. A linear regression between two datasets is displayed with a green dashed line. (a) All months in the period 2019-2022 are included. (b) Months that are included are those for which inferred emissions have been obtained with an averaging of 18 daily emissions or more. The color scale represents the number of days involved in the calculation of points.

The method can also be applied to a smaller area of the domain. The north-eastern part of the country is interesting in this respect: it is very sparsely populated and the human activity there is mainly associated with oil and gas activities and the production of electricity. In this region, three power plants (Ras Laffan A, B and C) have a total capacity of 3.35 GW, and are responsible of about 45 % of the total electricity generation throughout the year, as shown on Figure 11. These power plants are independent water and power plants (IWPPs), serving as both desalination plants and power plants. Other than these plants, the region also has gas liquefaction facilities, as Qatar is a major producer of LNG. Refineries are also present. In the life cycle of gas, only extraction and combustion are high NO$_x$-emitting processes: emissions from midstream (liquefaction and transport) and downstream (refining) activities are 20 to 30 times lower per unit of fuel (Marais et al., 2022). As the oil and gas extraction processes are located outside this area (mostly offshore), it can be considered that the majority of the NO$_x$ emitted in this region come from the combustion of gas in the power plants. This is not the case for other power plants in the country, which are located in urban areas where emissions from different sectors overlap. We focus on this area and calculate the corresponding monthly emissions. The three power plants are located within the same pixel, and the difficulty for TROPOMI to visualise this region with good quality measurements is less pronounced than for other regions identified in Section 5.3. Because Ras Laffan plants are not the only ones participating in the country's electricity generation, the use of load curves is not longer valid to infer total emissions on a given period, and only emissions at 13:30 can be considered. Assuming a correct calculation of emissions, the observed emissions display a Gaussian distribution around the power plants. Using a zonal cross-section of obtained emissions in a 28-km band centered around the mean latitude of the plants, we estimate the mean emissions by fitting a Gaussian curve $e_{\mathrm{NO_x}}(\lambda) = B + \frac{E_0}{\sigma\sqrt{2\pi}}\exp(-\frac{(\lambda-\lambda_0)^2}{2\sigma^2})$ on the profile, with $B$ the observed emissions above desert areas and seas, $\lambda_0$ the mean longitude of the power plants, $\sigma$ the standard deviation (measured as an angle) and $E_0$ the total emissions. $\lambda$ is the zonal angle. The best approximation is displayed on Figure 13.

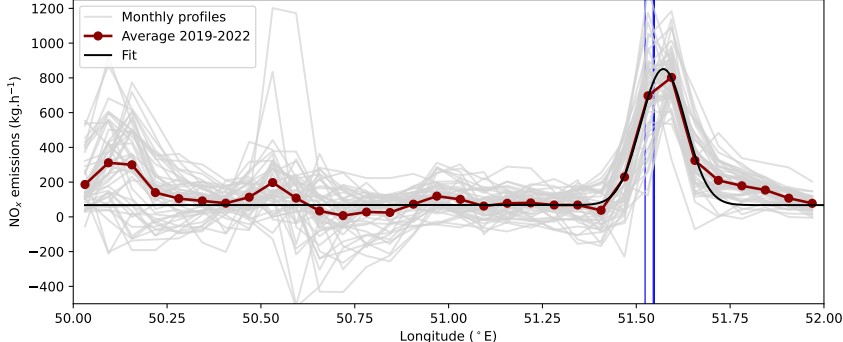

Figure 13: Zonal cross sections of 13:30 NO$_x$ emissions at the latitude of Ras Laffan power plants. The average profile for 2019-2022 is displayed in red, and the corresponding Gaussian fit in black. Individual monthly profiles are displayed in light grey. The positions of the three power plants are marked with vertical blue lines.



Monthly profiles peak around the location of the power plants whose height vary without displaying any particular seasonality. The fit of the averaged profile leads to a low positive background of 67.8 kg.h$^{-1}$ per unit of band surface, i.e. about 0.390 mg.m$^{-2}$.h$^{-1}$, and average emissions of 1.86 t.h$^{-1}$ for the three power plants. Two lower peaks are also identified in the west, corresponding to emissions from southern Bahrain and Dammam (Saudi Arabia).

Here, only a small part of the domain is considered, for which the power sector is by far the major contributor to NO$_x$ emissions. It is therefore not necessary to consider the seasonality of the other sectors in the estimation of an emission factor, as it was done earlier when the emissions of the whole domain were considered. However, as the scaling assumption can no longer be used, other assumptions must be made. One assumption would be that these plants serve as intermediate-load power plants: 13:30 being a peak of consumption at any time of the year, the Ras Laffan plants would be running very close to their maximum capacity, and the annual variability of power generation by the three plants observed in Figure 11 would be due to the decrease of power generation during low demand hours. This would be consistent with the lack of seasonality detected in the monthly profiles in Figure 13. In this case, comparing the emissions at 13:30 with the total capacity of the plants provides an emission factor of 0.557 t$_{NO_x}$.GWh$^{-1}$. As the total capacity is used in the calculation, this value must be considered as a lower limit estimate.

Another possible assumption is that the three plants are used as peak-load power plants: in this case, they would be used at 13:30 to respond to the daily peak. This use would increase with the height of the peak, and therefore with the temperature. In this case, the annual variability of power generation by the three plants would be due to their variable use at 13:30. The absence of a clear annual variability in our TROPOMI-inferred emissions would be explained by the large uncertainties in the calculation of our emissions, which concern only 6 pixels of our domain. Here, the large share of the three power plants in the electricity production seems to rule out a peak-load functioning. However, it is also possible that some of the plants correspond to different functions within the national grid.

## 6.2 Comparison with emission inventories

We compare TROPOMI-derived NO$_x$ emissions to the Emissions Database for Global Atmospheric Research (EDGARv6.1) for 2018 and the CAMS global anthropogenic emissions (CAMS-GLOB-ANT_v5.3) from 2019 to 2022. The two inventories provide 0.1°×0.1° sectorial gridded emissions on a monthly basis. After aggregating the different sectors of activity, CAMS-GLOB-ANT and EDGAR inventories directly provide the anthropogenic NO$_x$ emissions over the same domain. Both inventories display NO$_x$ emissions that are significantly higher than our estimates. According to CAMS-GLOB-ANT, the annual emissions for 2019, 2020, 2021 and 2022 are 164.8, 164.6, 165.3 and 166.4 kt respectively, whereas we only estimate emissions of 111.5, 111.8, 114.4 and 121.0 kt for these same years. According to EDGAR, NO$_x$ emissions are even higher, reaching 193.8 kt in 2018. It should be noted that the same version of EDGAR estimates emissions of 111.1 kt in 2007, which is similar to the reported terrestrial emissions for the same year, as well as our estimated emissions for 2019-2022.

The sectoral shares of emissions estimated by EDGAR and CAMS-GLOB-ANT differ. In EDGAR, the transport sector accounts for 42 to 54 % of the emissions, whereas it has a nearly constant share of 58 % in CAMS-GLOB-ANT. In the remaining emissions, industry accounts for about 2.8 kt per month in both inventories, which constitutes a lower share in EDGAR (between 13 and 23% of terrestrial emissions) than in CAMS-GLOB-ANT (about 18% of terrestrial emissions). Such shares are lower than reported shares for 2007, where the cement sector alone accounted for 23.4% of total emissions. Finally, power sector emissions in both inventories account for a significant share of total emissions, but with a higher value for EDGAR (35 % of the total budget for 2018) than for CAMS-GLOB-ANT (20 % of total emissions in 2019-2022). The two values are respectively higher and lower than the reported share for power emissions, which was estimated at 28.1 % for terrestrial emissions in 2007. Another major difference concerns the seasonnal cycle of those power emissions. In 2018, EDGARv6.1 shows a pronounced variability, with extremes of 7.0 kt emitted in May and 3.5 kt in December (see Supplementary Material). The ratio between these two values remains lower than the ratio between the extremes of the monthly electricity generations, which has varied between 2.2 and 2.8 over the last four years, but it highlights a difference with the previous version of EDGAR (EDGARv5.0) which showed almost no seasonality in power emissions. This lack of seasonality is also found in the emissions of CAMS-GLOB-ANT_v5.3, which uses EDGARv5.0 as a basis. We also note that while these seasonality differences between TROPOMI-inferred emissions and inventories were already highlighted for the previous versions of EDGAR and CAMS-GLOB-ANT in Egypt, the difference between mean annual NO$_x$ emissions was much lower (Rey-Pommier et al., 2022).

## 6.3 Emission factor for the power sector

The report for emissions in 2007 indicates an emission factor for the power sector of 0.392 t$_{NO_x}$.GWh$^{-1}$, and the corresponding power plants are still operating today. In the previous sections, we estimated an emission factor of 1.773 t$_{NO_x}$.GWh$^{-1}$ using all monthly emissions in the internal mask and assuming the power sector was the only source of variable NO$_x$ emissions. We




also estimated a value of 0.557 $t_{NO_x}.GWh^{-1}$ for Ras Laffan power plants using mean emissions at 13:30 under the hypothesis of a intermediate-load power plant functionning. To infer emission factors from inventory estimates, power emissions in EDGAR and CAMS-GLOB-ANT and total electricity generation provided by the Planning and Statistics Authority (PSA) between 2018 and 2022 can also be used to infer emission factors of 1.406 $t_{NO_x}.GWh^{-1}$ and 0.647 $t_{NO_x}.GWh^{-1}$ respectively. Table 1 summarises all the different emission factors that have been calculated. Even though EDGAR's emission factor is particularly high compared to other estimates, the five listed values are plausible, as the majority of gas-fired power plants have a $NO_x$ emission factor value between 0.1 and 10 $t_{NO_x}.GWh^{-1}$ (Miller, 2004).

| Year(s) | Source | Data for $NO_x$ emissions | Data for power | Method | Emission factor ($t_{NO_x}.GWh^{-1}$) |
|---|---|---|---|---|---|
| 2007 | All PPs | Reported emission factor (Initial National Communication to the UNFCCC) | | | 0.392 |
| 2018 | All PPs | EDGARv6.1 inventory $NO_x$ emissions | Annual power generation (PSA) | Ratio between total power emissions and electricity generation | 1.406 |
| 2019-2022 | All PPs | CAMS-GLOB-ANT_v5.3 inventory $NO_x$ emissions | Monthly power generation (PSA) | Ratio between total power emissions and electricity generation | 0.674 |
| 2019-2022 | Internal mask | Monthly TROPOMI-inferred $NO_x$ emissions | Monthly power generation (PSA) | Linear fit between monthly power emissions and electricity generation | 1.773 |
| 2019-2022 | Ras Laffan power plants | 13:30 TROPOMI-inferred $NO_x$ emissions | Power plant capacities (PSA) | Gaussian fit of zonal cross-section for Ras Laffan power plants | 0.557 |

Table 1: $NO_x$ emission factors for electricity production inferred from different sources and methods. The value is calculated from monthly emissions and electricity generation, fluxes at 13:30 and capacities or directly reported by Qatar authorities.

The comparison between these different emission factors should be made with caution. The factor reported by the report for 2007 emissions takes into account all of the power plants in the country, the oldest having been built in 1980. Moreover, some units of one of the Ras Laffan plants were built between 2010 and 2011, i.e. after the publication of the report. It is also possible that other power plants have been upgraded in the last years. Finally, the reported emission factor for 2007 accounts for the public electricity production and the water production, but the monthly reports from Kahramaa do not specify whether the electricity generation from the Ras Laffan power plants account for the large amount of electricity that is used in desalination processes (Kahramaa, 2023). Monthly reports show that the water production from the IWPPs in the country does not have a clear annual cycle, the production generally ranging between 50 and 60 millions of cubic meters per month in the entire country. The amount of electricity needed for this water production varies with the technologies used in the IWPPs, and accounts for about 20 % of the consumption in total electricity (Okonkwo et al., 2021). Consequently, if the electricity used for water production is not accounted for in the electricity generation for monthly reports, then all inferred emission factors are over-estimated.

## 7 Uncertainties and assessment of results

The long time series of average $NO_2$ concentrations in cities in several countries with similar economic and industrial developments (Dubai, Manama, Kuwait City) show a stagnation of $NO_x$ levels over the period 2005-2014 (Lelieveld et al., 2015). It is therefore not clear whether the $NO_x$ emissions of the country in 2019-2022 have increased from their value in 2007, as suggested by CAMS and EDGAR, or decreased, as our TROPOMI-based estimates seem to suggest. We don't know exactly how parameters in EDGARv6.1 and CAMS-GLOB-ANT_v5.3 are estimated in order to calculate $NO_x$ emissions, which means we cannot conduct a discussion on such bottom-up estimates accordingly. However, we can identify several factors that could influence of our top-down $NO_x$ emissions. The following aspects can be considered:

- As discussed in Section 5.3, a large portion of the territory, which includes some of the most emissive facilities in the country, is not visible by TROPOMI with high-quality data, and emissions from very emissive pixels might be underestimated. As this situation only occurs from 2019 to 2021, the magnitude of the induced underestimation is the difference between inferred emissions in 2022 and inferred emissions in 2019-2021, i.e. about 6 %.

- TROPOMI retrievals are sometimes biased due to poor estimation of the air mass factor or local effects under particular vertical distribution conditions (Griffin et al., 2019; Lorente et al., 2019; Judd et al., 2020; Wang et al., 2021). It should be noted that the latest versions of TROPOMI (v2.x) have tropospheric VCDs that are between 10 % and 40 % larger than the first versions (v1.x), depending on the level of pollution and season (Van Geffen et al., 2022). The chemical transport model TM5, which is used in the retrieval of the operational TROPOMI data, has been shown to underestimate surface level pollution while overestimating NO2 at higher levels above the sea (Latsch et al., 2023; Rieß et al., 2023). Such a bias is probably not negligible and probably accounts for significant part of the difference between our estimates and inventories, but also within our estimates, as surface albedo values differ between desert areas and urban zones.

- The vertical levels on which parameters in Equation 1 are estimated might be incorrect. Most studies use values





averaged within the PBL, which is higher in summer (Lorente et al., 2019; Lama et al., 2020), or averaged within a fixed vertical layer (Beirle et al., 2019). Here, we interpolated the fields of $\mathbf{w}$, $T$ and [OH] using the first pressure levels in ERA5 and CAMS at a mean pressure of 987.5 hPa. As Qatar is a flat country, this level corresponds to an altitude between 220 and 250 m. Because vertical transport of $NO_x$, which is emitted mainly from combustion engines and industrial stacks, is
generally minor compared to horizontal transport (Sun, 2022), $NO_x$ is confined to these first hundred metres above ground level. In Egypt, Rey-Pommier et al. (2022) have shown that the consideration of higher levels decrease mean temperature which tends to increase emissions, balanced by the decrease of mean OH concentrations with altitude which tends to decrease the sink. The net effect would be a decrease of emissions of a few percents during most of the year.

 - The $NO_x$-to-$NO_2$ ratio might locally be under-estimated. The conversion of NO to $NO_2$ by the reaction with $O_3$ is
balanced by the photolysis of $NO_2$ which reforms NO, leading to a stabilisation of the ratio a few kilometers downwind the source. The stationnary regime can also be displaced by the presence of volatile organic compounds. Near important sources of emissions, which mainly produce NO, a high $NO_x$-to-$NO_2$ ratio should be expected. We can therefore consider that for pixels containing hotspots close to their boundaries, such as power plants, it should be much higher than the chosen value of 1.32 (Hanrahan, 1999; Goldberg et al., 2022), leading to an underestimation of our emissions locally. With power plants
concentrating a large share of the $NO_x$ emissions in our study, this effect is probably not negligible and probably accounts for a large part of the difference between our estimates and inventories.

 - The sink term might be under-estimated or over-estimated. If the transport term is assumed to be estimated correctly, an underestimation of the sink term would lead to significant negative emissions in many parts of the domain, while an overestimation would lead to non-emitting areas with $NO_x$ emissions that are higher than noise. None of these effects seem
to appear for averaged emissions for the 2019-2022 period, but monthly maps show significant portions of the territory with negative emissions in winter and spring months, while summer months display abnormally high emissions over deserts and seas. Those effects are not visible for most of the autumn months. We can therefore assume that the sink term is slightly under-estimated in winter and spring months and over-estimated in summer months, due to errors in the estimation of OH in CAMS and/or the presence of additional sinks in summer. Accounting for this effect would not change the averaged
emissions for the period 2019-2022, but would reduced the seasonality observed on Figure 6. The emission factor calculated from emissions in the internal mask would also be reduced.

 These effects must be taken into account for understanding the differences between our model, reported emissions and inventory emissions. The uncertainties used here must also be considered. $NO_x$ emissions, as estimated with the flux-divergence method, are calculated with the use of several quantities. The uncertainty ranges shown in Figures 10 and 12 are
calculated from uncertainty statistics whose references are presented in this section, assuming that they correspond to standard deviations. The uncertainty of tropospheric $NO_2$ columns under polluted conditions is dominated by the sensitivity of satellite observations to air masses in the lower troposphere, expressed by the AMF, and the corresponding relative uncertainty is of the order of 30 % (Boersma et al., 2004), and is likely related to the a priori profiles used within the operational retrieval that do not reflect well the concentration peak of $NO_2$ near the ground. For the Middle East region, the impact of the a priori
profile is less critical, as surface albedo is generally high and cloud fractions are generally low. Thus, we do not expect a bias of this importance, and consider a relative uncertainty of 30 % for the tropospheric column as reasonable. For wind module, we assume an uncertainty of 3 m.s$^{-1}$ for both zonal and meridional wind components. For OH concentrations, the analysis of different methods conducted by Huijnen et al. (2019) showed smaller differences for low latitudes than for extratropics, but still significant. We thus take a relative uncertainty of 30 % for OH concentration. For the reaction rate $k_{\mathrm{mean}}$, we use
the value of the corresponding relative uncertainty, as estimated by Burkholder et al. (2020).

## 8  Conclusion

In this study, we investigated the potential of a top-down model of $NO_x$ emissions based on TROPOMI retrievals and a flux-divergence scheme applied at high resolution over Qatar for the last four years (2019-2022). This scheme requires different parameters to be calculated and consists in the calculation of a transport term that uses horizontal wind, and the calculation
of a sink term that requires temperature data and OH concentration to illustrate the chemical loss of $NO_x$. Results illustrate the difference between localised and diffuse sources of $NO_x$. For diffuse sources such as the Doha urban area, the transport term and the sink term similarly contribute to the total $NO_x$ budget, whereas the transport term is higher than the sink term for localised sources such as the gas power plants located in the north-east of the country. The emissions from other hotspots, such as cement plants in the west part of the country, could not be correctly estimated from 2019 to 2021 due
to unavailability of high-quality TROPOMI retrievals. Our estimated $NO_x$ emissions show a weekly variability which is consistent with the social norms of the country and an annual variability which is consistent with its electricity generation. In this study, estimated emissions are similar to reported emissions in 2007, but they are 1.44 times lower than emissions in the CAMS-GLOB-ANT_v5.3 inventory for 2019-2022 and 1.68 times lower than emissions in EDGARv6.1 for 2018. They have an annual cycle whose relative amplitude is higher than those two inventories. These notable differences may be subject



to further discussion regarding sectoral activity data and emission factors used in global inventories. Finally, this study is also an attempt to retrieve the $NO_x$ emission factor of the power sector. This top-down estimation is made possible by the desert features of Qatar, which allow to consider only one chemical sink, and also by the low diversity of its economy in which the power sector is the major source of variable $NO_x$. An emission factor is also estimated for a group of isolated gas power plants. The emission factors are 1.42 and 4.52 times higher than the value reported for 2007. Moreover, the emission factor of the entire power sector is higher than that of the three isolated gas power plants, indicating higher emission factors for other power plants in Doha. Although several limitations exist in the estimation of these results, we believe that such a calculation can be reproduced in other countries and for power plants that share similar characteristics. Overall, this study highlights the potential of TROPOMI to compensate for non-existent, inaccurate or outdated inventories by providing low-latency emissions estimates. The development of similar applications is likely to provide a better monitoring of global anthropogenic emissions, therefore helping countries to report their emissions of air pollutants and greenhouse gases as part of their strategies and obligations to tackle air pollution issues and climate change.

**Code availability.** Code will be made available on request.

**Data availability.** The TROPOMI Sentinel 5P Product Algorithm Laboratory (S5P-PAL) reprocessed data (processor version 2.3.1) from January 2019 to October 2020 has been used. The OFFL stream has been used afterwise, using processor versions 2.3.1 and 2.4.0 from November 2022. The TROPOMI $NO_2$ product is publicly available on the TROPOMI Open hub (http://www.tropomi.eu/data-products/data-access, TROPOMI Data Hub, 2022) while the S5P-PAL reprocessed data can be found on the S5P-PAL data portal (https://data-portal.s5p-pal.com, ESA, 2022). CAMS data can be downloaded from the Copernicus Climate Data Store (https://ads.atmosphere.copernicus.eu/cdsapp#!/dataset/cams-global-atmospheric-composition-forecasts, ECMWF, 2022a). The European Centre for Medium-Range Weather Forecasts (ECMWF) ERA5 reanalysis be downloaded from the Copernicus Climate Data Store (https://cds.climate.copernicus.eu/cdsapp#!/dataset/reanalysis-era5-pressure-levels-monthly-means, ECMWF, 2022b). Electricity generation data can be downloaded from The Planning and Statistics Authority Portal (https://www.psa.gov.qa/en/statistics1/Pages/default.aspx, last access: 31 January 2023), and from Kahramaa Portal (https://www.km.qa/MediaCenter/Pages/Publications.aspx, last access: 24 January 2023). Emissions reported for year 2007 is available on the UNFCCC website (https://unfccc.int/resource/docs/natc/qatnc1.pdf, last access: 26 August 2022). EDGARv6.1 emissions are provided by https://edgar.jrc.ec.europa.eu/emissions_data_and_maps. CAMS-GLOBANT_v5.3 emissions are available at https://eccad3.sedoo.fr.

**Author contributions.** AR analysed the data, prepared the main software code and wrote the paper. FC provided the TROPOMI $NO_2$ data product and corresponding gridded maps. ISB provided time series of electricity consumption. PC, TC, JK, and JS contributed to the improvement of the method and the interpretation of the results. All authors read and agreed on the published version of the paper.

**Financial support.** This study has been funded by the European Union's Horizon 2020 research and innovation programme under grant agreement N°856612 (EMME-CARE).

**Competing interests.** The authors declare that they have no conflict of interest.

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
