# Peer review of "Detecting nitrogen oxide emissions in Qatar and quantifying emission factors of gas-fired power plants – A four-years study"

_EGUsphere, 2023_

## Author Comment (AC1)

**Replies to comments provided by Anonymous Referee #1**
Title: Detecting nitrogen oxide emissions in Qatar and quantifying emission factors of gas-fired power plants - A four-years study
Author(s): Anthony Rey-Pommier et al.
https://egusphere.copernicus.org/preprints/2023/egusphere-2023-1024

*We would like to thank the reviewers for their careful reading, that led to interesting comments and improvements of the article. Minor results will be added in the revised versions of the manuscript and the Supplementary Material.*

**IMPORTANT**: We noticed a mistake in our estimation of power emissions in CAMS-GLOB-ANT due to a code typo. We corrected the mistake: power emissions in CAMS are now 1.81 times higher than they were in the manuscript, and the corresponding total emissions are now 1.16 times higher. The most significant change thus concerns one of the different estimations of the power plant emission factor in Section 6.3 and Table 1 which is now 1.219 $t_{NOx}$/GWh (previously 0.674 $t_{NOx}$/GWh). Results involving this inventory (lines 23, 454, 461, 463, 466, 484 and 572 in the current manuscript version) have been modified as a consequence, but the conclusions remain unchanged.

**Questions provided by Anonymous Referee #1**

This manuscript applies the flux divergence method to estimate NOx emissions over Qatar using TROPOMI NO2 retrievals. It represents an incremental development on the author's previous paper for emissions in Egypt. The paper is clearly written and appears to be thorough and sound. I am happy to recommend it for publication.

General Comments:

Urban emissions: as you note, Doha coincides with 5 gas power plants, making it difficult to separate emissions. However, it would be interesting to show estimated emissions of the urban and residential sectors versus the power and industrial sectors. These are readily available for EDGAR and CAMS. They would also improve the discussion of seasonal and day-of-week variability below. → In CAMS-GLOB-ANT, emissions from the different sectors do not show any seasonality, with the power, industry and transport sector at around 5.1, 2.8 and 8.0 kt/month respectively. On the other hand, those sectors in EDGAR show clear variations between months, especially the power sector. The following figure corresponds to Figure S4 but without stacking the emissions.

[Figure]

Figure AC1-1: Monthly $NO_x$ emissions by sector for Qatar for 2018 in EDGARv6.1

The industry emissions are slightly lower during summertime; they can illustrate the lower production due to the reduced outdoor activity mentioned in Section 2. Power emissions are lower

during wintertime, but the observed cycle has an amplitude much lower than what is observed in the electricity generation data. Finally, transport emissions do not vary much throughout the year. These elements are added to the revised version of the manuscript (Section 6.2). However, EDGAR and CAMS-GLOB-ANT inventories cannot be used in the discussion regarding the weekly cycle, since they both have a 1-month resolution.

Fig. 7: This makes me wonder if a coarse land-use mask is used. Work with the TROPOMI methane product found that a new high resolution water mask had to be used for coastal areas (de Foy et al., 2023). This problem is much more acute for methane than for NOx, but still it might have an effect here.

→ In the TROPOMI product, the parameter `surface_classification` in `NO2___/PRODUCT/SUPPORT_DATA/INPUT_DATA` is a combined land/water mask and surface classification data field. The water mask has been taken from Carroll et al. (2009), which has a 250 m resolution. We do not know exactly how the TROPOMI product uses this mask but the corresponding parameter does not have a coarse resolution for both versions, although there is a noticeable improvement from version 2.3.1 to version 2.4.0 according to the following figure:

[Figure]

Figure AC1-2: Example of the use for the land/water mask used in the TROPOMI L2 NO2 product for versions 2.3.1 (left, 2021/07/24) and 2.4.0 (right, 2022/07/24).

Fig. 8: I wonder if you could show boxplots here to get a sense of the difference as a function of the variability. I think you did a sum of the flux divergence over the whole of Qatar? What happens if you look at different areas? I would expect a stronger weekday effect over residential area, and a weaker one over power plants and industrial facilities. As a check, I think it would be good to show the weekly cycle in VCD as well as in flux divergence.

→ We prefer not to use boxplots here to focus on highlighting the absence of significant differences between years. Indeed, the week-end effect is stronger over residential areas and weaker over power plants. The figure below displays the weekly cycle of $NO_2$ VCDs, OH concentrations and $NO_x$ emissions over the whole country, the Ras Laffan power plants and the urban area of Doha for the 2019-2022 period (at around 13:30 LT). It will be added in the revised Supplementary Material. OH concentrations do not vary much within the week (in the CAMS product, we do not know if a weekly variation was introduced in the calculation of OH), so the observed trend is mainly due to the observed VCDs in the calculation of emissions.

[Figure]

Figure AC1-3: Mean weekly profiles for NO$_2$ tropospheric vertical columns, OH concentration and NO$_x$ emissions for the entire country (left), the Ras Laffan power plants in the north (middle), and the Greater Doha area (right). 2019-2022 averages are given and represented by the 100% line on the y-axis.

Note that the 4-year average is calculated without accounting for values lower than the 5$^{th}$ percentile or higher than the 95$^{th}$ percentile (as explained in Section 5.4), and that the re-scaling using the load curve is not used. Moreover, the power plant emissions are calculated summing emissions of the 6 pixels that are the closest to the power plants. For those three reasons, the mean values shown on this figure do not correspond exactly to those of Figures 10, 12 and 13.

Fig. 10: Maybe in SI you could show the monthly variation, or at least put color bars over the summer months to help see the annual cycle. In the text you say there is no seasonal signal in the VCD. I think it would be good to show the cycle in VCD as well as flux divergence side by side (as for the weekly cycle). Given the large seasonal cycle in electricity cycle, a lack of cycle in the TROPOMI results suggests that something else is going on. For example transport and industrial emission may be stable throughout the year.

→ Colors have been added to Figure 10 to visualise the annual cycle in NO$_x$ emissions (green: MAM; red: JJA; yellow: SON; blue: DJF). Concerning the VCD cycle, it could be considered inconsistent to put on a same graph emissions based on pixel sums within a mask and VCD columns when the extent of NO$_2$ plumes can go multiple kilometres beyond the mask limits, as shown on Figure 2. However, it is possible to calculate the mean VCD value over different hotspots, using only the closest pixels (where NO$_2$ is maximum) inside the mask. Doing so, we obtain the following figure for the mean VCD over the Ras Laffan power plants (6 pixels), the cement plants in the west (4 pixels) and the Greater Doha area (15 pixels):

[Figure]

Figure AC1-4: Time series for mean NO$_2$ tropospheric VCDs above Ras Laffan power plants in the north, cement plants in the west, and the greater Doha area in the east.

Emissions from the cement plants in the west and from the gas power plants in the north do not show any particular seasonality. It is not the case for the greater Doha area, with lower VCD values during summer than during fall and winter. This cycle is in phase opposition to that of $NO_x$ emissions shown on Figure 10, because the emissions cycle reflects more OH concentration variations (i.e. lifetime variations) than the $NO_2$ column budget. There is a non-linear relationship between OH and $NO_2$: following Valin et al. (2011), $NO_2$ levels over the main emitters in Qatar (higher than $2.0 \times 10^{15}$ molecules/cm$^2$) are such that a linear increase in $NO_2$ levels is linked to an exponential decrease in OH concentrations. This relationship, which highlights a dominance of the OH variation with respect to the $NO_2$ variation in the high-$NO_2$ regime, might explain why the calculated emissions cycle is in phase opposition to that of the TROPOMI $NO_2$ VCDs. These elements are now mentioned in the revised version of the manuscript (Section 5.5).

Getting actual emission totals from the flux divergence method involves uncertainties, especially due to lifetime as you note. It would be interesting to see how your method compares to the values reported in the global catalog (Beirle et al., 2021). It would also be interesting to see how your evaluation of TROPOMI and EDGAR sources compares with that reported for large point sources and urban areas in South Asia (de Foy et al., 2022).

→ The Ras Laffan power plants are absent in the first version of the catalog by Beirle et al. (2021). However, they are present in the improved version of the catalog which has been published a few days ago (Beirle et al., 2023), with emissions estimated at $1.81 \pm 0.37$ t$_{NOx}$/h, which is very close to our value of 1.86 t$_{NOx}$/h estimated in Section 6.1. If we compare of TROPOMI-derived $NO_x$ emissions to those of de Foy et al. (2022), we observe that our results are notably higher: in South Asia, isolated gas power plants are Rohini and Faridabad near Delhi and Sheikhupura near Lahore, and the corresponding estimated emissions are 2.7, 4.4 and 5.4 times lower than those of the Ras Laffan complex, with capacities 2.3, 4.8 and 14.9 times lower, suggesting similar fuel efficiencies for Ras Laffan, Rohini and Faridabad and a lower efficiency for Sheikhupura compared to that of Ras Laffan. The two articles are mentioned in the revised version of the manuscript (Section 6.1).

Minor Comments:

Fig. 12: I think you are plotting one point per month, with total NOx emissions and Electricity generation over the whole of Qatar? I think the explanation could be clearer to help the casual reader.

→ The label of the Figure has been changed to "Comparison between monthly TROPOMI-derived $NO_x$ emissions for the entire Qatar territory and corresponding electricity generation according to Planning and Statistics Authority reports. [...]"

Line 311: replace "estimaed" with "estimated".
→ Done.
* * *
References:

Beirle, S., Borger, C., Dörner, S., Li, A., Hu, Z., Liu, F., ... & Wagner, T. (2019). Pinpointing nitrogen oxide emissions from space. *Science advances*, *5*(11), eaax9800.

Beirle, S., Borger, C., Jost, A., & Wagner, T. (2023). Improved catalog of NOx point source emissions (version 2). *Earth System Science Data*, *15*(7), 3051-3073.

Carroll, M. L., Townshend, J. R., DiMiceli, C. M., Noojipady, P., & Sohlberg, R. A. (2009). A new global raster water mask at 250 m resolution. *International Journal of Digital Earth*, *2*(4), 291-308.

Valin, L. C., Russell, A. R., Hudman, R. C., & Cohen, R. C. (2011). Effects of model resolution on the interpretation of satellite NO2 observations. *Atmospheric Chemistry and Physics*, *11*(22), 11647-11655.

---

## Author Comment (AC2)

**Replies to comments provided by Anonymous Referee #2**
Title: Detecting nitrogen oxide emissions in Qatar and quantifying emission factors of gas-fired power plants - A four-years study
Author(s): Anthony Rey-Pommier et al.
https://egusphere.copernicus.org/preprints/2023/egusphere-2023-1024

*We would like to thank the reviewers for their careful reading, that led to interesting comments and improvements of the article. Minor results will be added in the revised version of the manuscript and the Supplementary Material.*

**IMPORTANT**: We noticed a mistake in our estimation of power emissions in CAMS-GLOB-ANT due to a code typo. We corrected the mistake: power emissions in CAMS are now 1.81 times higher than they were in the manuscript, and the corresponding total emissions are now 1.16 times higher. The most significant change thus concerns one of the different estimations of the power plant emission factor in Section 6.3 and Table 1 which is now 1.219 $t_{NOx}$/GWh (previously 0.674 $t_{NOx}$/GWh). Results involving this inventory (lines 23, 454, 461, 463, 466, 484 and 572 in the current manuscript version) have been modified as a consequence, but the conclusions remain unchanged.

**Questions provided by Anonymous Referee #2**

The authors infer NOx emissions in Qatar using satellite NO2 observations and compare it with the bottom-up inventories. It is well written. The results look sound. I recommend publication after minor revision.

General comments:

Section 4.2. The divergence method used here has been proposed by existing studies, e.g., Beirle et al. (2011). I think the authors shall give the credit to those studies by clarifying that this study is an application of an existing method. How the method is different (if any) from existing studies shall be highlighted.
→ The divergence method follows Beirle et al. (2019) (not Beirle et al., (2011)). The article is mentioned but we recognize we don't emphasize enough on the importance of this first study and the differences it has with our method. Section 4.2 is modified accordingly. The following sentences are modified/added:
- "As a second step, we derive top-down $NO_2$ production maps with the flux-divergence method, which has originally been proposed by Beirle et al. (2019)." *(modified)*
- Through the OH concentration, we enable a variability in the chemical lifetime. This variability is not allowed in the original version of the first-divergence method by Beirle et al. (2019), which relied on heavy averaging over time to infer emissions at the scale of cities and power plants. Here, seasonal and spatial variations of lifetimes are resolved, thus limiting the errors in the estimation of the daily sink term. Although errors remain high when estimating daily emissions, averaged monthly emissions are correctly resolved above the main emitters. *(added)*

The uncertainty of using 5 percentiles as background shall be discussed.
→ The main reason why the 5[th] percentile of external mask pixels is chosen is because we want to estimate the background using pixels that do not contain anthropogenic $NO_2$. The domain is quite small and most of it is polluted with emissions from either Qatar, Bahrain, Saudi Arabia, UAE and shipping/flaring emissions in the Persian Gulf. The entire domain has 1536 pixels and the external mask has 1065 pixels, so the 5[th] percentile-threshold should correspond to the 53 lowest pixels. To

illustrate the evolution of the inferred background with respect to the percentile chosen, I compared the daily background value obtained with the 10th, 15th and 20th percentiles, as well as the pixels involved in their calculation for year 2022 (counted as the number of times a pixel is lower than the background value during the year):

[Figure]

Figure AC2-1: Time series of estimated background using different percentiles of $NO_2$ columns in the external mask for year 2022.

[Figure]

Figure AC2-2: Frequency map of $NO_2$ columns below a given percentile (from left to right: 5th, 10th, 15th and 20th percentile) used in the estimation of the background for year 2022.

Using a background defined as a higher percentile involves pixels that are increasingly close to the main emitters. On average, the difference between the 5th percentile and percentiles 10, 15 and 20 are $0.13\pm0.04\times10^{15}$ molecules/cm$^2$, $0.22\pm0.06\times10^{15}$ molecules/cm$^2$, $0.31\pm0.08\times10^{15}$ molecules/cm$^2$, which is small compared to the total tropospheric VCDs which is often above $4.0\times10^{15}$ molecules/cm$^2$ for the Ras Laffan power plants and $4.0\times10^{15}$ molecules/cm$^2$ for Greater Doha. In terms of $NO_x$ emissions, using the 20th percentile instead of the 5th, the size of the internal mask, and a mean lifetime of 4.3 hours (average 2019-2022 over the domain), would result in an average lowering of the inferred emissions of ~0.6 kt/month, i.e. about 6% of the average monthly emissions for 2019-2022.

Section 6.1. I understand the correlation between emissions and generation data is relatively low for monthly data. How is the correlation compared with that between bottom-up estimates and generation data? The comparison could help explain the inconsistency between TROPOMI-derived emissions and generation data.
→ $NO_x$ emissions estimates from EDGAR and CAMS-GLOB-ANT do not correlate with electricity generation data. Electricity generation is shown to be highly variable, whereas the variability in total emissions from these two inventories is very low in comparison. If considered only the power emissions, EDGAR has some variability, but it is lower than that observed in electricity generation

and the profile is slightly different. The following table shows the correlation coefficient ($R^2$) between electricity generation data and bottom-up total and power emissions, with TROPOMI-derived emissions as comparison.

| Dataset compared with monthly power generation 2019-2022 | $R^2$ |
|---|---|
| Total emissions – TROPOMI (2019-2022) | 0.400 |
| Total emissions – TROPOMI – only more than 18 days in average (2019-2022) | 0.657 |
| Total emissions – EDGARv6.1 (2018) | 0.062 |
| Power emissions – EDGARv6.1 (2018) | 0.117 |
| Total emissions – CAMS-GLOB-ANT_v5.3 (2019-2022) | 0.001 |
| Power emissions – CAMS-GLOB-ANT_v5.3 (2019-2022) | 0.137 |

Table AC2-1: Correlation coefficient of the comparison between inferred or inventory total or power $NO_x$ emissions and electricity generation data.

The absence of correlation between electricity generation and $NO_x$ emissions in inventories are briefly mentioned in the revised version of the manuscript (Section 6.2).

Specific comments:

Line 13. Regularly updated.
→ Done.

Line 19. No dash in under-estimated.
→ We corrected the spelling (and "over-estimated" as well) but it seems like both spellings are correct, depending on the dictionary. There might be a subtlety we do not get.

Line 65. The sentence is too long to read.
→ The sentence "Because the incomplete combustion of hydrocarbons produces $NO_x$, the exploitation of such oil and gas resources is a source of air pollution: as a consequence, the transport sector is a source of emissions, as well as the power sector, which is dominated in Qatar by gas power plants." has been replaced by "The exploitation of such oil and gas resources is a source of air pollution, due to $NO_x$ emissions during the incomplete combustion of hydrocarbons. The power sector, which is dominated by gas power plants, as well as the transport sector, are thus important contributors to the $NO_x$ levels throughout the country."

Line 169. Covered?
→ The sentence has been slightly changed: "In satellite retrievals, the $NO_2$ signal from a sparsely populated area or a small industrial facility may be difficult to identify due to high noise levels or natural emissions."

Is there any specific reason for choosing 30 km/h as the criteria to remove high-wind speed days?
→ The 30 km/h value has been chosen as a threshold because it corresponds to the minimal value for the wind module to reach the closest high emitters of Qatar. Manama and the cement plants in the east (angle ~ -75°) are separated by ~ 110 km. Manama and the Ras Laffan power plants (angle ~ -15°) are separated by ~ 105 km. The average lifetime value calculated from CAMS OH in the area between the two countries is about 3.5 hours. The corresponding value for minimal wind module is therefore ~ 105 km / 3.5 h ≈ 30 km/h, which is why we have chosen this value as a threshold.

It should be noted that a further analysis has been conducted after reading this comment. Indeed, the lifetime in the concerned regions vary. On average during the 2019-2022 period, it reaches 5.2 hours during winter months (DJF). During wintertime, a plume originating from Manama could thus theoretically reach the power plants in Ras Laffan with a module of ~ 100 km / 5.2 h ≈ 21 km/h, which is under the 30 km/h threshold. However, three points must be made:

- This situation should not appear frequently throughout the year: the MAM, JJA and SON periods have average lifetimes of about 3.1, 2.1 and 3.2 h, which is too low to correspond to a wind module lower than 30 km/h.
- In practice, an analysis of the wind in the region shows that the wind angle during wintertime rarely corresponds to the Manama – Ras Laffan direction, but often corresponds to direction between Manama and the cement plants in the west of Qatar (Figure 2 is actually an example of this) which is unfortunately rarely observed by TROPOMI (at least before version 2.4.0).
- Unrealistic negative emissions frequently appear on maps in the region between Bahrain and the west of Qatar, which might indicate an underestimation of the sink term through an overestimation of the lifetime.

After using lower values at 25 and 20 km/h, we observed that discarded days went from 169 to 240 and 309 respectively. About half of the additional discarded days correspond to days between December and March included, for which 4.25 additional days are discarded on average. Lowering the threshold generally leads to a decrease in emissions. On average, this does not impact the value of main emissions, since the absolute change in total $NO_x$ emissions with a threshold of 20 km/h is about -2.6% on average in 2019-2022, and about -6.0% on average for months between December and March included. Months for which the absolute change is higher than 10% are January 2019 (-10.7%, 3 additional discarded days), December 2019 (-20.1%, 7 additional discarded days), February 2021 (-14.2%, 7 additional discarded days) and December 2022 (-10.7%, 3 additional discarded days). These large diminutions are mostly explained by the fact that the additional discarded days included days for which pixels above the cement plants and Doha were visible, increasing thus the number of pixels which are never observed within a month.

With such comparisons, it can be concluded that lowering the threshold below 30 km/h in order to avoid overestimating emissions through to the inclusion of pollution from Bahrain would only be appropriate for the winter months. For these months, although a threshold of 20 or 25 km/h would be more appropriate, the impact on total emissions is marginal and only lead to a slight reduction in emissions. The months for which the reduction in $NO_x$ emissions is significant are months for which the lowering of the threshold leads to the omission of highly emissive areas due to the discarding of some TROPOMI images in the average, which means that the reduction in total emissions is not due to the effect that the threshold lowering was intended to avoid. Most of this discussion is added in the revised Supplementary Material.